# Antibacterial Adhesion Strategy for Dental Titanium Implant Surfaces: From Mechanisms to Application

**DOI:** 10.3390/jfb13040169

**Published:** 2022-09-29

**Authors:** Jingwei Yu, Minghao Zhou, Luxuan Zhang, Hongbo Wei

**Affiliations:** State Key Laboratory of Military Stomatology, National Clinical Research Center for Oral Diseases & Shaanxi Engineering Research Center for Dental Materials and Advanced Manufacture, Department of Oral Implants, School of Stomatology, The Fourth Military Medical University, Xi’an 710032, China

**Keywords:** antibacterial, titanium implant, coating, surface nano-topographies

## Abstract

Dental implants are widely used to restore missing teeth because of their stability and comfort characteristics. Peri-implant infection may lead to implant failure and other profound consequences. It is believed that peri-implantitis is closely related to the formation of biofilms, which are difficult to remove once formed. Therefore, endowing titanium implants with anti-adhesion properties is an effective method to prevent peri-implant infection. Moreover, anti-adhesion strategies for titanium implant surfaces are critical steps for resisting bacterial adherence. This article reviews the process of bacterial adhesion, the material properties that may affect the process, and the anti-adhesion strategies that have been proven effective and promising in practice. This article intends to be a reference for further improvement of the antibacterial adhesion strategy in clinical application and for related research on titanium implant surfaces.

## 1. Introduction

Dental implants are widely used to restore missing teeth because of their stability and comfort [1]. Due to its biocompatibility, titanium and its alloys are suitable materials for dental implant technology. Titanium has been the most extensively used material for dental implants [2]. However, the incidence rates of subsequent peri-implant mucositis and peri-implantitis, which may lead to implant failure, top 43% (confidence interval: 32–54%) and 22% (confidence interval: 14–30%), respectively [3]. Clinically, the infection is often controlled by mechanical removal, laser treatment, and local or systemic medication [4]. However, marginal bone loss around the implant may occur if the infection is uncontrolled, and peripheral reflection can be seen on X-rays. Due to infection and bone loss, the osseointegration of the implant starts to fail, and the implant must be removed as soon as possible [4].

It is considered that bacterial infection and related immune–inflammatory responses are the main causes of peri-implant mucositis and peri-implantitis [5,6]. The half desmosomes and basement membranes of the implant–epithelial interface are incomplete; the tissue around implants lacks periodontium at the tooth–bone interface; the surrounding collagen fibers are placed parallel to the implant; and vessels providing nutrition are limited [7]. These factors make soft tissue seal poor and prone to infection by bacteria. Furthermore, the number of Langerhans cells in the mucosa around the implant is lower than that in normal tissue [8]. Innate immunity is decreased, so the likelihood of infection increases significantly. In the first two weeks, the wound is directly exposed to the oral environment, so it has more contact with bacteria [9,10]. It is at elevated risk for early infection, and this may affect healing. After bacterial entry, the mucosa produces a protective stress reaction, and various immune-related signaling pathways are activated [11,12,13,14,15]. Macrophages [16], neutrophils [17], B cells, and T cells [18] increase. As a result of inherent immunity and adaptive immunity, bacteria and foreign materials are cleaned up. Local titanium ions (Ti) [17,19,20], occlusion [21], and systemic factors (such as smoking [22] and diabetes [23]) all influence this process. It is normal for organisms to be able to regulate themselves to achieve balance. However, if bacteria are difficult to eradicate or if other factors trigger inflammation, this will result in peri-implant inflammatory diseases and tissue destruction [24,25,26].

Oral microflora can form obstinate biofilms. Biofilms are organized bacterial populations surrounded by extracellular macromolecules, adhering to the surface of living or unliving objects. It is difficult for the immune system or antimicrobial agents to remove biofilms completely [27,28,29,30]. Routine treatments, such as mechanical treatments, cannot completely remove biofilms. All these factors mean that, once biofilms mature, they can remain for a long time. Biofilms are considered to be the initiating factor of peri-implantitis [31]. Scientists have conducted many studies to determine how to efficiently remove biofilms [32,33,34]. Preventing biofilm formation is the most important way to avoid peri-implantitis and implant failure [35]. In addition to aseptic techniques and prophylactic antibiotics, antibacterial coatings have received considerable attention. Antibacterial coatings on titanium surfaces can be divided into two types: passive coatings and active coatings [36]. Passive coatings on titanium (Ti) mainly kill the bacteria on contact, but they do not kill the plankton or bacteria that dwell in the bone tissue around the Ti implant. Active coatings mainly involve the release of antibacterial agents to kill the bacteria, but this may result in the development of bacterial resistance [37,38,39].

Recently, due to the widespread abuse of antibiotics [40], the frequent emergence of drug-resistant bacteria [40], the cytotoxicity of bactericidal substances [41], and the difficulty in re-supplying drugs and in removing carriers after drug release, researchers have started to investigate antibacterial adhesion. Here, bacterial adhesion refers to the process of free bacteria adhering to the surface of dental implants and growing. The aim of this review is to briefly introduce the process of bacterial adhesion in the oral cavity, to explore how implant surface properties affect this process, and to summarize the current antibacterial strategies targeting bacterial adhesion.

We searched the PubMed and Web of Science. The keywords and Boolean operators are (implant AND antibacterial AND adhesion) OR adhesion.

## 2. The Adhesion Process of Oral Bacteria

Bacteria that initially colonize the surface of teeth or implants during biofilm formation are called pioneer bacteria. The adhesion process is complex and regulated by multiple factors. It can generally be simplified into two steps: approach and attachment [42,43,44,45,46]. The process is shown in Figure 1.

Approach refers to the proximity to a surface. Bacteria can approach surfaces via passive or active movement. The flow in bulk liquid, Brownian motion, and gravitational forces generate passive movement [47,48]. The active movement of bacteria involves swimming toward a surface through their flagella [49,50,51]. In bulk liquids, bacteria can move freely due to the bacterial load, but the flow and shear rates significantly affect this process [52]. In the vicinity of a surface, active movement plays a major role. By physically and chemically detecting the presence of a surface, bacteria can find the right direction [44]. Chemical detection includes sensing the concentration of hydrogen ions, antibacterial agents, and some biological signaling molecules [44]. In this way, bacteria can move toward surfaces with a specific nature. Physical detection includes receiving physical signals with the fimbriae or flagella and obtaining information through signal transduction [53]. When flagellar rotation is impeded, bacteria can also perceive a surface [53]. The deformation of the cell membrane caused by a surface also leads to the transmission of signals [46,54,55,56]. As a result, bacteria can swim toward the surface.

Attachment refers to the generation of a permanent type of adhesion. Bacteria attach to surfaces through three types of interactions [46]: nonspecific physical–chemical interactions, specific interactions, and surface mechano-sensing. They work sequentially in this process. Nonspecific interactions include a variety of non-covalent interactions, such as van der Waals interactions, electrostatic interactions, and acid-based hydrophobic interactions. They are often affected by surface properties and environmental factors [42,44,57,58]. In the attachment process, two minima exist for the interaction-free energy [48,59]. The first minimum is at a separation of 10 nm, and the second is at a separation of 1 nm. The energy barrier is of a few kT. This means that they need other interactions to overcome this barrier. Specific interactions involve the paired binding of multiple adhesins to receptors [46]. Saliva coats the implant due to the oral environment. There are proteins and sugars that provide the sites for specific interactions [60,61]. Some bacteria have a network of long polysaccharide chains and other biopolymers on their cytoderm called the polymeric brush layer. As a result, they can produce steric forces, which play an important role in attachment [42,62]. They have been demonstrated to be more important than van der Waals and electrostatic forces [62]. Surface mechano-sensing is the process through which bacteria actively regulate themselves through signal transduction after they meet suitable surfaces [46]. Bacterial appendages are the main providers. In this process, the flagellum not only functions as an adhesin but also explores the surface topography and accesses microenvironments inaccessible to the whole cell in order to increase contact [63,64,65]. Fimbriae, curli, and pili also have receptors that bind to specific or wide ranges of nonspecific substrates involved in surface attachment [49,66,67]. The bacteria reposition their cells and surface structures to achieve permanent attachment and to produce adhesin molecules. Nonspecific interactions are reversible processes, while the following specific interactions and surface mechano-sensing are firmer and irreversible [68,69]. Bacteria activate specific genes in this irreversible phase to create a protective extracellular matrix in order to resist immunity, antibiotics, and harsh conditions. This is why subsequent treatment is difficult [69]. Interestingly, bacteria do not turn on their genes until they are firmly attached to the surface [70]. This makes antibacterial attachment an opportunity to fight infection.

After colonizing, the pioneer bacteria modify the micro-environment and provide binding sites. These sites lead to aggregation and co-aggregation. With the production of extracellular polymeric substances, the biofilm matrix progress builds up, and larger bacterial aggregates develop. Further remodeling and maturation form a microcolony [71]. Then, biofilm dispersal can occur, and bacteria return to a planktonic lifestyle [72]. Phenol-soluble modulins (PSMs) have an important role in this phase [73]. The detaching bacteria can adhere to other surfaces, spreading infection [74]. Eventually, the infection becomes chronic and difficult to treat [75].

## 3. Implant Surface Properties Affecting Bacterial Adhesion

Due to the complexity of bacterial adhesion, the factors influencing it are also diverse: pH, oxygen, bacterial species, and multiple strains. Here, we focus mainly on the properties of implant surfaces. Their mechanisms of influence on bacterial adhesion overlap, and they describe surface properties from different perspectives. They are described below in order of importance.

### 3.1. Roughness and Surface Topography

Roughness is one of the main factors affecting the adhesion of pioneer bacteria [76]. It is generally accepted that there is a threshold for the effect of roughness [76,77,78,79]. When the roughness is less than 0.21 μm, it has no significant effect. When the roughness exceeds 0.21 μm, it plays an important role [80]. The adhesive area increases with the roughness. Moreover, the furrows on a surface provide a barrier for bacteria to resist shear forces. Consequently, the amount of adhered bacteria increases, and the biofilm matures faster [81,82]. Nevertheless, bacterial species and experimental conditions affect the outcome. Bacteria cannot simply be inhibited from adhering to surfaces, nor is there a universally optimal roughness to prevent them from adhering to surfaces [83].

Roughness describes the ridges or projections of a material surface, which is usually defined as the small distance between two peaks or two troughs (wave distance). With the development of nanotechnology, roughness is no longer relevant to describing nano-scale surfaces. Based on numerous studies, nano-scale morphology affects regular bacterial growth, causing bactericidal effects [83,84]. Some natural nanostructures, such as lotus leaves, have a definite impact on bacteria, as they are superhydrophobic [85] and can resist bacterial adhesion. Cicada wings exhibit superhydrophobicity, and their nano-pillars can damage the bacterial cell membrane to kill bacteria [86,87,88,89]. Gecko feet obtain wetness resistance through bristles and have antimicrobial properties to a certain extent [90,91,92,93].

### 3.2. Hydrophilicity

When the hydrophilicity of a material decreases, bacteria can adhere to the surface through hydrophobic interactions. When it increases, a layer of water film forms on the surface and reduces the adhesion of the bacteria. Simultaneously, it is more conducive to the adhesion of osteocytes [76,94,95]. Nonetheless, some surfaces, such as lotus leaves, have waxy layered micro/nano structures that can trap air and achieve the effect of superhydrophobicity, making bacteria adhere loosely [96,97]. After the trapped air dissolves, this topography promotes the adsorption of nonspecific proteins and enhances the adhesion of bacteria [98].

### 3.3. Charge 

Most bacteria are negatively charged, so negatively charged materials are less likely to be adhered to [99]. The structure can also be influenced by surface charges [100]. As a material becomes more hydrophilic, so does the effect of the charge on the bacteria. Due to the oral environment, calcium ions may play an important role in promoting bacterial adhesion. Similarly, negatively charged polymers, such as heparin, can promote biofilm formation, similar to extracellular DNA secreted by Staphylococcus aureus [101].

Some cationic groups, such as quaternary ammonium and polyethyleneimines, have antibacterial activity. They can also affect the long-term structure of biofilms [102]. Concurrently, shear forces in the oral cavity can remove dead bacteria, which is conducive to maintaining antibacterial properties. However, the presence of salivary films can affect its properties, and dental materials may also affect the composition and properties of salivary films [103,104,105,106]. Further investigations of its properties in a real-world oral environment are necessary.

### 3.4. Surface Free Energy

A higher surface free energy generally corresponds to more surface-active ions, resulting in stronger attraction, so it is generally believed that a lower surface free energy may reduce bacterial adhesion [58,82]. A lower surface free energy can make biofilms relatively immature. For some nanostructures, a higher nano-roughness increases the surface energy and promotes protein and cell adsorption. Meanwhile, a lower nano-roughness can reduce the anchor points for bacteria, thus reducing adhesion [107]. It is also believed that regular nanostructures may impact bacterial adhesion more than irregular nanostructures [107].

The above properties are summarized in Table 1. In addition to these factors, other properties, such as the stiffness of the material, also affect bacterial adhesion [108]; however, very few studies have been conducted on this mechanism [109].

## 4. Anti-Adhesion Strategies for Titanium Implants

The process of infection and healing around dental implants is often described as a “race” between bacteria and cells [110], and anti-adhesion is one of the common strategies used to inhibit bacterial growth. Routine anti-adhesion strategies include the use of anti-adhesion coatings and anti-adhesion nano-topographies (Figure 2) [111,112,113,114,115,116]. Anti-adhesion coatings rely primarily on the nature of their materials to reduce various interactions. The strategy of anti-adhesion nano-topographies is to artificially build a nano-morphology in order to give surfaces an anti-adhesive quality. Some cases also use other materials to form coatings. These cases still use the nano-morphology strategy because the anti-adhesive quality comes from their surface morphology. Researchers use coatings because their materials are suitable for creating nano-scale morphologies. 

### 4.1. Anti-Adhesion Coating

Currently, researchers are studying the hydrophilicity of materials [117,118,119]. A firm hydration layer can form a physical barrier to prevent bacteria from approaching it. Furthermore, hydrogen bonds and electrostatic interactions exist between material particles and water molecules. Bacteria must destroy the original connection to replace the water molecules and form new force interactions (van der Waals and electrostatic forces) with material surfaces. This process forms a thermodynamic energy barrier. The hydrophilicity is related to the polymer structure, particularly the effect of carbon spacer lengths (CSLs) [120]. Meanwhile, an in vitro study has shown that this process does not affect cell adhesion, possibly due to the stronger peptidoglycan in the bacterial cell wall and the smaller bacterial size [121]. The main goal of studying anti-adhesive coatings is to find suitable materials with a strong hydration layer [122]. In addition, an antibacterial adhesion coating can also become multifunctional by adding other components. The following is introduced depending on whether the function is single or multiple [111,112,113,114,115,116,123].

#### 4.1.1. Simple Anti-Adhesion Coatings

Polyethylene glycol (PEG) coatings are the most common anti-adhesive coating [111,124,125]. PEG is highly hydrophilic, and its special structure can link water molecules through hydrogen bonds [126], forming a hydration layer [127]. Combined with the PEG coating, the hydration layer forms a barrier in space to prevent bacteria from approaching. The mobility and repulsion of PEG further enhance the barrier [124]. Even though PEG is chemically stable, it is easily oxidized in most biochemically relevant solutions and does not provide long-term antibacterial protection after implantation. Moreover, PEG in vivo may cause dysregulation in the immune–inflammatory reaction, complicating the surrounding tissue immunological environment and affecting osseointegration [128,129].

A variety of coatings, such as zwitterionic polymer coatings [130], chitosan coatings [131], and hyaluronic acid coatings [132], can also inhibit bacteria adhesion through their hydration layers. Among these polymers, zwitterionic polymers have recently received a great deal of attention. Zwitterionic polymers usually form side chains with positive/negative ion functional groups on the main chain and can also polymerize molecules with positive and negative ions in a 1:1 ratio [133,134,135,136,137], and a suitably formulated mixed-charged zwitterionic copolymer can achieve a better antibiofilm effect than that of some classical zwitterionic polymers [138]. The solvation of zwitterionic polymers can produce high solvent water retention, and dipole–electrostatic interactions and the electrostatic induction of ions can form a firm hydration layer [139]. Moreover, zwitterionic polymers have strong salt affinities, so electrolytes in physiological environments enter their molecules. As a result, zwitterionic polymers stretch, and ionic solvation occurs, enhancing hydrogen bonds. This allows them to occupy more solvent water and to improve their hydrophilicity [140]. Compared with PEG coatings, zwitterionic coatings are more stable and more firmly hydrated [141,142,143,144]. The water molecules around zwitterionic polymer coatings are arranged in different directions; thus, they have higher degrees of freedom and an energy barrier. As a result, they have a better antibacterial effect [145]. Additionally, this promotes template biomineralization and osseointegration [146,147]. 

Figure 3 shows more information about simple anti-adhesion coatings [129,148,149,150].

In addition, UV-irradiated titanium dioxide is super-hydrophilic and can achieve antibacterial effects [154]. Titanium nitride (TiN) can also inhibit bacteria adhesion due to its high chemical inertness, hardness, low friction coefficient, and corrosion resistance, significantly reducing the interactions [127,155].

In addition to traditional anti-adhesion coatings, quorum-sensing coatings are also gradually attracting the attention of scientists. Quorum sensing refers to the phenomenon in which bacteria release signal molecules, causing them to increase in number, affect the expression of specific genes and behavioral responses, and regulate their physiological characteristics [156]. Currently, some substances, such as cinnamaldehyde, can interfere with quorum sensing and prevent biofilm formation [148]. However, because the specific process of quorum sensing varies across different bacteria, it is difficult to select a substance that can interfere with the quorum-sensing process of most bacteria, which brings great challenges to this pathway [149].

#### 4.1.2. Composite Anti-Adhesion Coatings

The effect of using the above anti-adhesion coatings alone is limited, and achieving the ideal antibacterial effect is difficult. Anti-adhesion coatings are often used in combination with systemic antibiotics [150]. To improve their performance, researchers have chosen to functionalize them further. Some studies have demonstrated that being loaded with antimicrobial peptides, antibiotics, bactericidal agents, or some metal ions can enhance their antibacterial properties [119,157,158]. Researchers previously grafted an antimicrobial peptide, Magainin I (Mag), to a titanium oxide surface [159]. PEG was also used to decrease adhesion. The adhesion of proteins and bacteria was considerably reduced in this coating. Due to this special coating, adherent bacteria grew slowly [159].

Osseointegration is another important process to be considered. Because, to some extent, bacterial adhesion and cell adhesion are similar processes, some anti-adhesion surfaces may simultaneously damage cell adhesion and affect osseointegration. Therefore, the coating often immobilizes bioactive growth factors and cell adhesion sequences to promote osseointegration. Additionally, some researchers have tried to segment antimicrobial peptides in these coatings, and the results are promising. For example, poly (l-lysine)-grafted-poly(ethylene glycol) (PLL-g-PEG) can decrease the adhesion of fibroblastic cells, osteoblastic cells, and bacteria, whereas PLL-g-PEG functionalized with peptides of the RGD (Arg-Asp-Gly) type can restore the adhesion of fibroblastic and osteoblastic cells [160]. This minimizes the effect of nonspecific protein adsorption on bacterial adhesion. As a result, RGD enhances osseointegration. Researchers have also tried other methods. The study conducted by Harris et al. covalently grafted dopamine, followed by carboxymethyl chitosan (CMCS) or hyaluronic acid-catechol (HAC). Then, vascular endothelial growth factor (VEGF) was conjugated to the polysaccharide-grafted surface. Its antibacterial properties and ability to promote bone cell growth were certified [161]. Loading antibacterial agents on bone induction surfaces is also a hot topic in current research [162].

These two objectives can also be achieved together, and the layer-by-layer (LAL) method is suitable [163]. More information is shown in Figure 4. 

### 4.2. Anti-Adhesion Nano-Topographies

With the development of nanotechnology, researchers have found many nano-topographies with antibacterial and even bactericidal properties, and they have begun to focus on specific surface topographies. Some natural nano-topographies and their properties are described in Table 2.

The most widely known anti-adhesive nano-topography is lotus leaves. The wax on lotus leaves has an infiltration angle of up to 161° through its chemical properties and unique layered micro/nano topography [85]. Two-dimensional nanoporous surfaces, three-dimensional nanotubule-like surfaces [174], and hydrophilic titanium dioxide nanotubes [175,176] can inhibit bacterial adhesion by reducing the surface area and trapping air to form superhydrophobic surfaces, but their antibacterial ability is limited. Some researchers have obtained a superhydrophobic surface on titanium surfaces by mimicking lotus leaves using femtosecond laser ablation technology, which has a significant inhibitory effect on the adhesion of Pseudomonas aeruginosa but an insignificant effect on Staphylococcus aureus. The authors believe that this may be because the spherical Staphylococcus aureus requires smaller adhesion points [177,178].

Additionally, the trapped air in the nanostructure gradually dissolves over time, leading to decreased antibacterial properties and even an increased adsorption of nonspecific proteins [98]. This is an important influencing factor in its antibacterial life. Moreover, since nanomaterials are still mostly tested in vitro, it is still difficult to determine their toxicity [179].

Cicada wings and single-wall carbon nanotubes have mechanical sterilization functions. The more complex and sharper the nanostructure, the stronger the bactericidal effect. In addition to titanium nano-morphologies, polymer, graphene, zeolite, and metal nanoparticles are gaining increasing attention from researchers. They can be loaded with antibacterial agents, metal ions, and bioactive factors to improve the antibacterial properties and enhance osseointegration. By doing so, implants become more stable. One example is shown in Figure 5 [180]. Another direction is to use nanoparticles as “vaccines”. If “smart” nanoparticles are designed to mimic bacteria in morphology and function, they can diagnose and treat peri-implantitis through immune-modulating mechanisms [181].

Wear and corrosion are also important factors in immune–inflammatory responses [182,183]. Frictional stresses may result in the production of pro-inflammatory cytokines, which are associated with the apoptotic response. Harmful corrosion products can also result in cytotoxicity and cause damage to tissues and organs. Titanium and its alloys have a lower hardness than other metals, so their wear hardness is lower [184,185]. Electrochemical corrasion can also easily occur due to the body fluid environment [186]. There are studies showing that specific polymer composite coatings and some metal nanomaterials can improve the wear and corrosion resistance [187,188,189,190,191]. More relevant studies are still needed, especially for these functional implant surfaces. 

Although these strategies have been explored for many years, studies in vivo are few, studies about clinical performance are few, and not one has reached the commercial stage [192]. Tetsurou et al. focused on the antibacterial effect of silver nanoparticles and studied them in vivo [193]. The result of the clinical trial proved that they can prevent biofilm accumulation. However, the mental cytotoxic issue, without an exact solution, limits the clinical application [194]. The complexities of the coating design, the poor or lack of antimicrobial strategies, and the safety issues in human clinical trials are the gap between laboratorial studies and clinical application [193].

Additionally, there is a lack of unified evaluation criteria for anti-adhesion strategies. Currently, most research is conducted in vitro. Researchers generally select the main bacteria causing peri-implantitis, such as *Escherichia coli, Staphylococcus aureus, Streptococcus mutans,* and *Porphyromonas gingivalis* [180,195,196], or they directly extract bacteria from saliva to simulate anaerobic and aerobic experiments. To demonstrate the antibacterial properties, they use the ratio of alive to dead bacteria calculated by immunofluorescence staining [193,197,198,199]. A selection of specific bacteria cannot replicate the complex microecological environment in the oral cavity, and there are no standard patients who can provide saliva, resulting in poor comparisons between experiments. The comparability between experiments may be improved if a unified composite strain and concentration are used based on the composition of human saliva and if the experiments are conducted under predetermined conditions.

## 5. Outlook

The current strategies for antibacterial adhesion of dental implants mainly focus on the hydrophilicity and nano-topography of the implant material to reduce bacterial adhesion and mitigate the risk of infection before biofilm formation. Although the anti-adhesion strategy avoids the risks of drug resistance and cytotoxicity, it still has many problems. As it has no active bactericidal function, once the bacteria are successfully colonized, they often cause infection [200]. Due to some common mechanisms in the adhesion process between cells and bacteria, some anti-adhesive coatings can also affect osseointegration and the healing effect of implantation [113]. In addition to the inability to remove coatings, the coatings themselves age and are difficult to reload. Even though fungicides and bioactive factors can improve some problems, peri-implant infection remains a problem.

There is a growing interest in antibacterial surfaces that are “smart” [200,201]. They remain “biologically inert” without making contact with bacteria to facilitate cell adhesion. Although they can efficiently kill bacteria and remove the accumulated dead bacteria after making contact with bacteria or triggering control conditions, they can maintain antibacterial properties for a long time [202]. Researchers are still far from finding a suitable antibacterial surface for clinical use. The ideal strategy should have good antibacterial properties that do not affect cell adhesion in vitro. Then, it should be proven to be secure, stable, and effective in vivo. The manufacturing method should be widely used so that the implant can be produced on a large scale. Additionally, the cost needs to be considered.

## Figures and Tables

**Figure 1 jfb-13-00169-f001:**
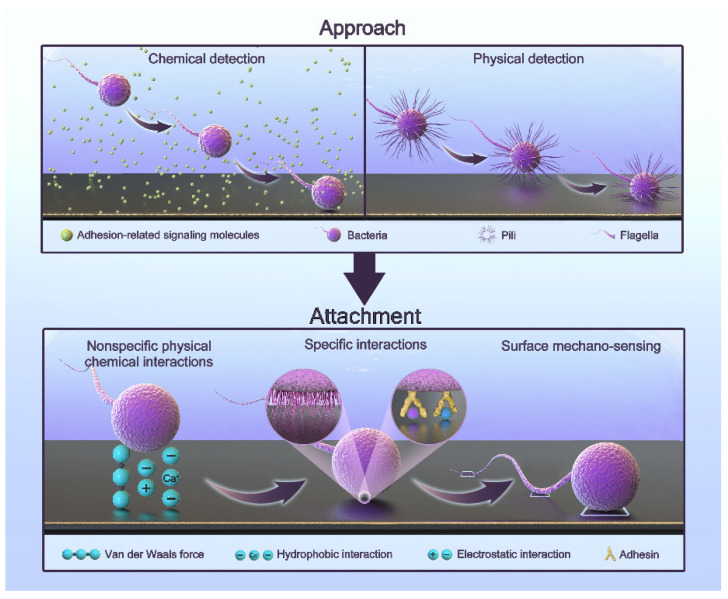
The adhesion process of oral bacteria.

**Figure 2 jfb-13-00169-f002:**
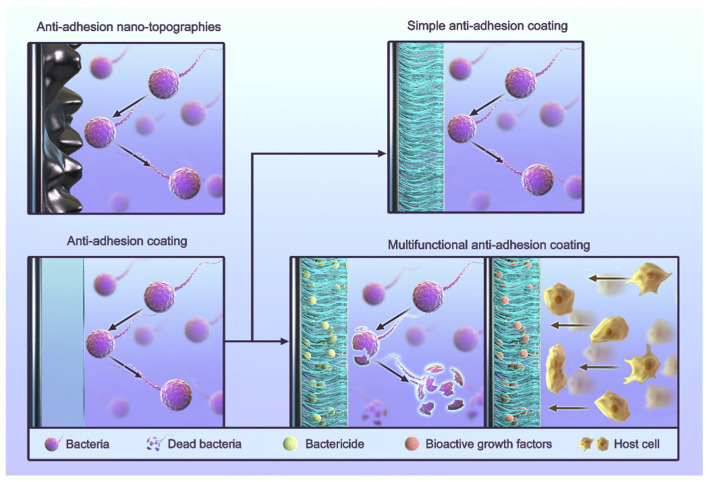
The classification of anti-adhesion strategies for titanium implants.

**Figure 3 jfb-13-00169-f003:**
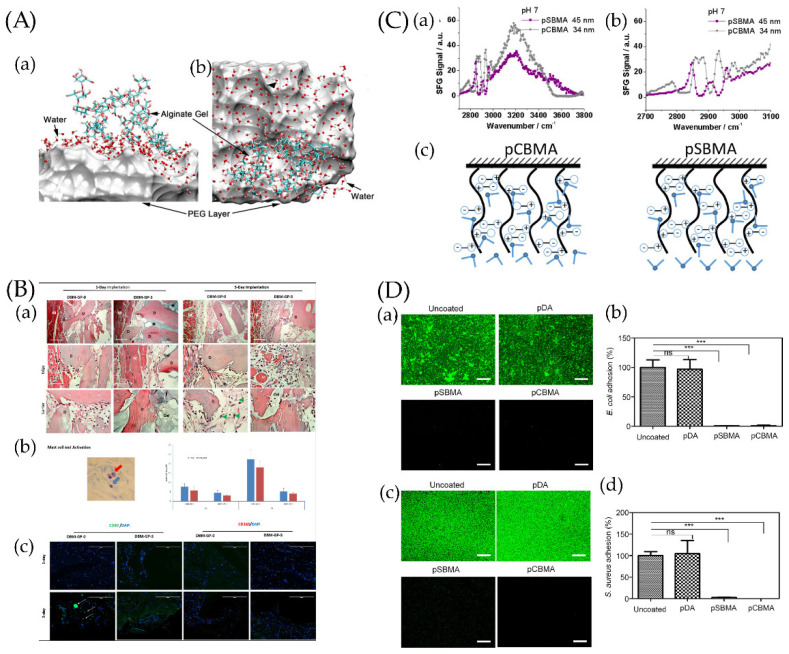
(**A**) Snapshot of a hydration water layer and alginate gel near a PEG coating: (**a**) side view and (**b**) top view. The PEG coating is represented by a van der Waals surface for clarity. Ref. [151] Reprinted with permission from Xiang et al. Copyright© 2016 American Chemical Society. (**B**) The resulting blocks contained 0, 20, 40, or 60 mg of PEG per gram of demineralized bone matrix (DBM) and were called DBMP-GP-0, DBM-GP-1, DBM-GP-2, and DBM-GP-3, respectively. Early-phase tissue reactions to PEG-cross-linked-gelatin-coated DBM. (**a**) DBM-GP-0 and DBM-GP-3 were explanted at days 1 and 3 postsurgery and H&E stained. D, DBM; M, muscle; gel, PEG cross-linked gelatin; mag, 10× and 40×. (**b**) Left, a representative image of mast cells (blue arrow) and their activation (red arrow); right, quantitative number of total and activated mast cells were measured at 1 and 3 days between DBM-GP-0 and DBM-GP-3. (**c**) IHC staining of CD80+ and CD163+ cells on explants. Scale bar = 100 μm [129]. Reprinted with permission from Bo et al. Copyright © 2020 American Chemical Society. (**C**) (**a**) Sum frequency generation (SFG) spectra collected from poly-(carboxybetaine methacrylate) (pCBMA)/water (pH~7) and poly-(sulfobetaine methacrylate) (pSBMA)/water (pH~7) interfaces. (**b**) Enlarged SFG spectra collected from pCBMA/water (pH~7) and pSBMA/water (pH~7) interfaces in the C−H stretching frequency region. (**c**) Scheme showing water molecules on two zwitterionic polymer surfaces [152]. Reprinted with permission from Xiaofeng et al. Copyright © 2019 American Chemical Society. (**D**) Antibacterial effect of pSBMA- and pCBMA-grafted Ti/TiO_2_ substrates against *E. coli* and *S. aureus*. Uncoated and polydopamine (pDA)-coated substrates were used as controls. (**a**,**c**) Representative fluorescence images and (**b**,**d**) quantification of E. coli and S. aureus cells adhering to surfaces after incubation for 6 h as determined by the SYTO 9/PI stain. The scale bars represent 200 μm. Each value represents the average and standard deviation of three different locations on three replicate samples. The fluorescence intensity of each sample was calculated as the percentage of the uncoated substrate. Statistical analysis was carried out using GraphPad Prism version 5. Each substrate was compared with the uncoated control using the unpaired Student’s *t* test (*** *p* < 0.0001, ns: not significant (*p* > 0.05)) [153]. Reprinted with permission from Yohan et al. Copyright © 2022 American Chemical Society.

**Figure 4 jfb-13-00169-f004:**
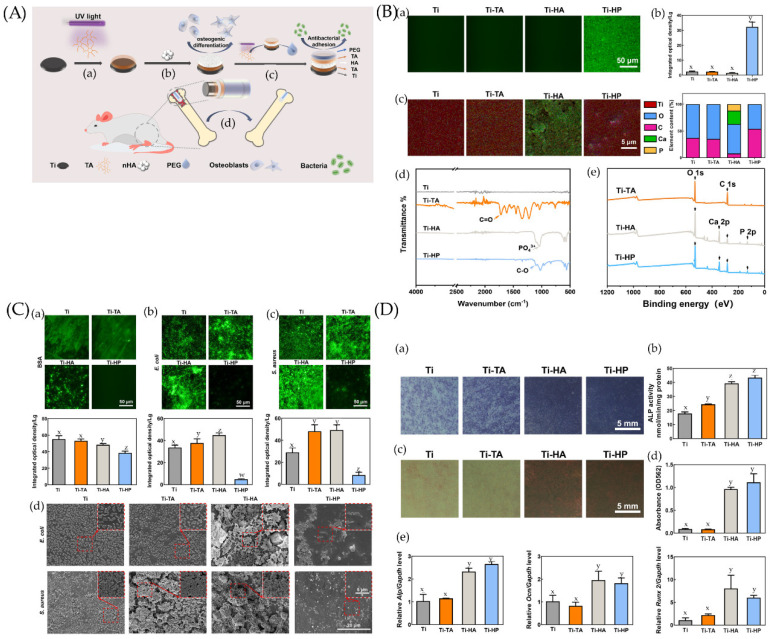
(**A**) Preparation and biological function of Ti-HP. (**a**) Tannic acid (TA) is deposited and polymerized on a Ti surface under UV irradiation. (**b**) A hydroxyapatite (HA) layer adheres to the TA layer and endows coating with osteogenesis ability. (**c**) The outermost PEG layer clinging to the superimposed TA layer enables the coating to inhibit bacterial adhesions. (**d**) The Ti-HP rod promotes osseointegration in a rat model. (**B**) Chemical characterization and surface wettability of coatings. (**a**) Fluorescence micrographs and (**b**) the corresponding integrated optical density (IOD) values of samples before and after being coated by polyethylene glycolated isothiocyanate (PEG-FITC). (**c**) The element mappings (Ti, O, C, Ca, and P) and quantified element analysis (O, C, Ca, and P) of the corresponding areas of Ti, Ti-TA, Ti-HA, and Ti-HP. (**d**) Attenuated Total Reflection–Fourier Transform Infrared Spectroscopy (ATR-FTIR) spectra of Ti, Ti-TA, Ti-HA, and Ti-HP. (**e**) X-ray photoelectron spectrometer (XPS) survey of the different steps in the preparation process. Dissimilar letters (x, y, z, and w) indicate values that are significantly different from each other’s group (*p* < 0.05). (**C**) Comparison of anti-adhesion properties among Ti, Ti-TA, Ti-HA, and Ti-HP. The fluorescence micrograph and the corresponding IOD values of (**a**) bovine serum albumin (BSA), (**b**) *Escherichia coli* (*E. coli*), and (**c**) *Staphylococcus aureus* (*S. aureus*) on surfaces after 24 h incubation. (**d**) Scanning electron microscopy (SEM) images of bacteria on samples incubated for 24 h. Dissimilar letters (x, y, z, and w) indicate values that are significantly different from each other’s group (*p* < 0.05). (**D**) Osteogenic differentiation of BMSCs on Ti, Ti-TA, Ti-HA, and Ti-HP. (**a**,**b**) Alkaline phosphatase (ALP) staining and quantitative analysis of bone marrow stromal cells (BMSCs) on samples after a 7-day osteogenic induction. (**c**) Alizarin Red S staining and (**d**) quantitative analysis among samples after a 14-day osteogenic induction. (**e**) mRNA expression levels of Alp, Ocn, and Runx2 determined by qRT-PCR. Dissimilar letters (x, y, z, and w) indicate values that are significantly different from each other’s group (*p* < 0.05). Ref. [163] Reprinted with permission from Yifang et al. Copyright © 2022 American Chemical Society.

**Figure 5 jfb-13-00169-f005:**
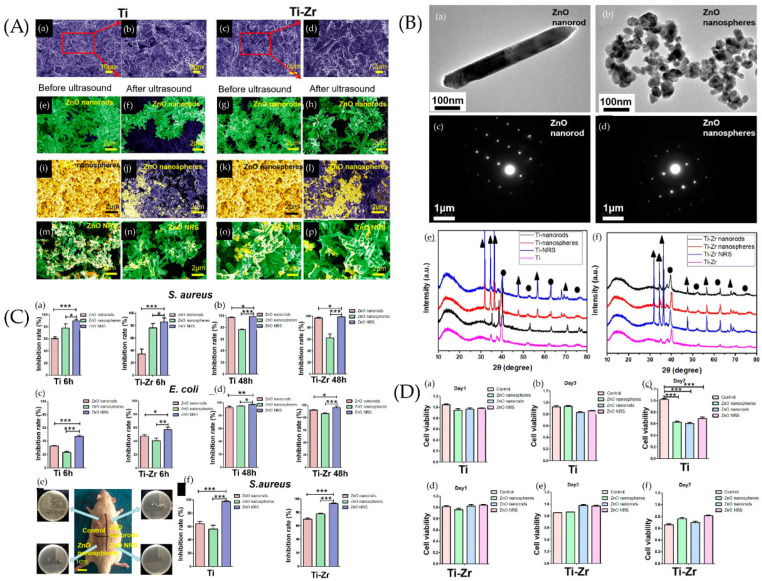
(**A**) SEM images of samples. SEM images of Ti and Ti−Zr slices were observed after sand blasting (**a**,**c**), where panels (**b**,**d**) are larger views of the red areas in panels (**a**,**c**). SEM images of ZnO nanorods on the surfaces of Ti slices (**e**,**f**) and Ti−Zr slices (**g**,**h**) were studied before and after ultrasound. SEM images of ZnO nanospheres on the surfaces of Ti slices (**i**,**j**) and Ti−Zr slices (**k**,**l**) were studied before and after ultrasound. SEM images of ZnO NRS on the surfaces of Ti slices (**m**,**n**) and Ti−Zr slices (**o**,**p**) were observed before and after ultrasound. (**B**) Transmission electron microscopy (TEM) and selected area electron diffraction (SAED) diffraction patterns of samples. TEM and SAED diffraction patterns of two different ZnO samples, (**a**,**c**) ZnO nanorods, and (**b**,**d**) ZnO nanospheres. X-ray diffraction (XRD) spectra of the Ti and Ti−Zr substrates coated with three different ZnO nanostructures (**e**,**f**) (▲ represents ZnO, and ● represents Ti). (**C**) Antibacterial effect of different ZnO samples against S. aureus and E. coli under different incubation times in vitro. ZnO samples cocultured with bacteria for 6 h (**a**,**c**) and 24 h (**b**,**d**). (**e**) Diagram of SD rats 2 weeks after implant surgery. Four respective plate colony counting photos are provided to show the amount of bacteria on the surfaces of implants. (**f**) Antibacterial capability of the different ZnO coatings on implant surfaces (Ti or Ti−Zr) against S. aureus 2 weeks after surgery. * *p* > 0.05, ** 0.01 < *p* < 0.05, and *** 0.001 < *p* < 0.01. (**D**) Cell viability of the human fibroblasts. The cell viability of the human fibroblasts in Ti (**a**–**c**) and Ti−Zr (**d**–**f**) groups after 1, 3, and 7 days of cell culture with leaching liquor of different samples is shown. *** 0.001 < *p* < 0.01). Ref. [180] Reprinted with permission from Xiaheng et al. Copyright © 2020 American Chemical Society.

**Table 1 jfb-13-00169-t001:** Summary of related factors and their effects.

Factors	Methods	Favorable Results	References
Roughness	Increases the adhesive area Provides a barrier against shear forces	Low roughness	[80]
Hydrophilicity	Forms a hydration layer	High hydrophilicity	[76,94,95]
Charge	Forms electrostatic interactions	Negative charge	[100]
Surface free energy	Provides an attractive force	Low surface free energy	[58,82]

**Table 2 jfb-13-00169-t002:** Natural nano-topographies and their properties.

Surfaces	Characteristics	Nano-Topographies	References
Taro leaves	Anti-biofouling, hydrophobic, and self-cleaning	Microscale elliptical bumps (10–30 µm in diameter) covered by hierarchal, waxy nano-scale epicuticular crystals	[164,165]
Lotus leaves	Anti-biofouling, hydrophobic, and self-cleaning	Micro-scale elliptical bumps, covered by nano-scale crystals	[85,165]
Shark skin	Self-cleaning, anti-biofouling, hydrophobic, drag-reducing, and aerodynamic	Triangular or placoid micro-structured riblets, some of which have small grooves in the direction of water scales	[166,167,168]
Gecko skin	Adhesion properties, anti-wetting properties, and bactericidal ability	A periodic array of hierarchal microscale keratinous hairs, approximately 30–130 µm in length, 5 µm in diameter, and split into hundreds of nano-scale spatula 200–500 nm in diameter	[169]
Cicada wing	Hydrophobic and bactericidal ability	Nano-pillar diameter range of 82–148 nm, 44–177 nm pillar spacing, and 159–146 nm in height	[88,170]
Dragonfly wing	Hydrophobic, self-cleaning, and bactericidal ability	Irregularly shaped nanostructures between 83.3 and 195 nm	[171]
Butterfly wing	Anisotropic flow effects, hydrophobic, low drag, anti-biofouling, and low bacterial adhesion properties	An array of aligned scales covered by hierarchal micro-grooves, approximately 1–2 µm in diameter	[172,173]

## Data Availability

Not applicable.

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
