# Peer review of "Antibacterial Adhesion Strategy for Dental Titanium Implant Surfaces: From Mechanisms to Application"

_jfb, 2022, doi:10.3390/jfb13040169_

Round 1

Reviewer 1 Report

The review article is interesting and well written. However, some changes are needed to increase its value. 

My suggestions and comments are as follows:

Please link the affiliations correctly to the authors names, using numbers, not crosses.

Line 17. Please consider replacing "will be" with intends to be or similar.

Line 22. "Dental implants are widely used to restore missing teeth because of their stability, comfort, and aesthetics." Sorry, it's misleading, please be more specific.

Line 23. "The biocompatibility of titanium and its alloy provides suitable materials for dental implant technology."  This is not an accurate expression, please rephrase.

Line 27. "CI". Please give full term for every abbreviation used.

Line 50,63,75,171,176,195-202,218. Reference needed.

Line 54. "Mechanical scaling cannot completely remove biofilms, too." Which is the relevance of this idea in the context?

Line 68. Consider replacing "This review will" with the aim of this review. 

Please add a sentence describing how the research was carried out, the databases searched, the key words used and the Boolean operators used. 

Line 78. Figure 1 caption. I presume it's bacteria, not bacterial.

Line 135. "we describe". Please use third person, instead of second. Same line 430.

Line 138,139,141,406. References should not be written in italics.

Line 156. I think hydrophilicity should replace hydrophilic. Same in Table 1.

Line 184. "Some researchers" and only one reference. Please correct.

Line 190. "very few studies" and no reference. The statement should be backed by references. Same for line 206. "Currently, researchers are studying the hydrophilicity of materials."

Line 212. An in vitro study instead of "in  vitro study" would be better.

Line 284.  "people" should be replaced with scientists or similar.

Line 295-297. References needed.

Line 312. "Their study". Who's study? Please specify.

Line 318. It's objects or objectives? Please verify.

Please highlight the changes in the revised manuscript and give lines in your point by point answer. Avoid "it has been corrected" or " done".

Author Response

Response to Reviewer 1 Comments

Point 1:Please link the affiliations correctly to the authors names, using numbers, not crosses.

Response 1: Thanks for your suggestion. It is our negligence. The right link is necessary. We have corrected to use the number to link the affiliations in Line 4. We have also add the description of authors’ contribution in Line 8.

Point 2: Line 17. Please consider replacing "will be" with intends to be or similar.

Response 2: Thanks for your comment. We are sorry for this wrong tense, and simple present tense is needed here. We have replaced “will be” with “intends to be” in Line 18. Now, the sentense is “This article intends to be a reference for further improvement of the antibacterial adhesion strategy in clinical application and related research on titanium implant surfaces”.

Point 3: Line 22. "Dental implants are widely used to restore missing teeth because of their stability, comfort, and aesthetics." Sorry, it's misleading, please be more specific.

Response 3: Thank you for your suggestion. It is our mistake.The aesthetics characteristics are mainly linked to the crown, and dental implants are widely used due to their stability and comfort. We have deleted the description of aesthetics in Line 24.

Point 4: Line 23. "The biocompatibility of titanium and its alloy provides suitable materials for dental implant technology." This is not an accurate expression, please rephrase.

Response 4: Thank you for your comment. We are very sorry for this confusing expression. Suitable materials are titanium and its alloy, and their biocompatibility is the key porperty. We have rephrased in Line 26.

Point 5: Line 27. "CI". Please give full term for every abbreviation used.

Response 5: Thanks for your comment. We are very sorry for our negligence. “CI” stands for confidence interval, and cannot be used directly. We have given full term for “CI” in Line 29.

Point 6: Line 50,63,75,171,176,195-202,218. Reference needed.

Response 6: Thanks for your reminding. We are very sorry for our negligence.We have added references in Line 53,68,83,178,184,204,226.

Point 7: Line 54. "Mechanical scaling cannot completely remove biofilms, too." Which is the relevance of this idea in the context?

Response 7: Thank you for your suggestion. We are sorry for the confusing context. Mechanical scaling is just a treatment. We want to use it to represent the routine treatments, and explain the difficulty to remove mature biofilms. We have added the relevant description in Line 59.

Point 8: Line 68. Consider replacing "This review will" with the aim of this review. 

Response 8:Thanks for your comment.We are sorry for this ambiguous expression. We have replaced “This review will” with “The aim of this review is to” in Line 74. Now the sentense is “Routine treatments, such as mechanical, cannot completely remove biofilms. All these mean that once biofilms mature, they can remain for a long time”.

Point 9:Please add a sentence describing how the research was carried out, the databases searched, the key words used and the Boolean operators used. 

Response 9: Thanks for your suggestion. It is meaningful for a review. We searched the PubMed and Web of Science. The key words and Boolean operators are (implant AND antibacterial AND adhesion) OR adhesion. We have added the description in Line 78.

Point 10: Line 78. Figure 1 caption. I presume it's bacteria, not bacterial.

Response 10:Thank you for your comment. We are very sorry for our negligence. Figure 1 describes the adhesion process of oral bacteria. We have replaced “bacterial” with “bacteria” in Line 85.

Point 11: Line 135. "we describe". Please use third person, instead of second. Same line 430.

Response 11:Thanks for your comment.We are very sorry for our negligence. It is unsuitable to be written in the first person.We have replaced “we” with “they” in Line 142.

Point 12:Line 138,139,141,406. References should not be written in italics.

Response 12: Thanks for your comment. We are very sorry for our negligence. We have removed italics in Line 145,146,148.

Point 13: Line 156. I think hydrophilicity should replace hydrophilic. Same in Table 1.

Response 13:Thanks for your comment. We are very sorry for our negligence. A noun is needed here according to the context. Same in Table 1. We have replaced hydrophilic with hydrophilicity in Line 163 and Table 1.

Comment 14: Line 184. "Some researchers" and only one reference. Please correct.

Response 14:Thank you for your comment. We are very sorry for our negligence. This is the result of one research, the expression of “Some researchers” is inaccurate. We have rephrased in Line 193.

Point 15: Line 190. "very few studies" and no reference. The statement should be backed by references. Same for line 206. "Currently, researchers are studying the hydrophilicity of materials."

Response 15:Thank you for your comment. We are very sorry for our negligence. We have added references in Line 198,214.

Point 16: Line 212. An in vitro study instead of "in vitro study" would be better.

Response 16:Thank you for your comment. We are very sorry for our negligence. A noun is needed here sa a subject. We have replaced “in vitro study” with “an in vitro study” in Line 221.

Point 17: Line 284.  "people" should be replaced with scientists or similar.

Response 17:Thanks for your suggestment. We are very sorry for the ambiguous description. We have replaced “people” with “scientists” in Line 292.

Point 18:Line 295-297. References needed.

Response 18:Thanks for your comment. We are very sorry for our negligence. We have added the references in Line 305.

Point 19:Line 312. "Their study". Who's study? Please specify.

Response 19:Thanks for your comment. We are very sorry for this inaccurate description. Harris et al. did this experiment. We have specified the scientists in Line 321.

Point 20:Line 318. It's objects or objectives? Please verify.

Response 20: Thank you for your comment. We are very sorry for our negligence. We have replaced “objects” with “objectives” in Line 326.

Reviewer 2 Report

Dear Authors,

I congratulate with you for the article. It is well written and it covers an interest topic.

Here my review:

Abstract : ''Dental implants are widely used to restore missing teeth because of their stability, comfort,  and aesthetic characteristics''.  Aesthetic characteristics are not linked to dental implants, but rather with the crown that rehabilitates the implant. Please reformulated to avoid misunderstanding. Same with the first phrase of the introduction.

The article well covers the aspects of the bacteria colonization and the anti bacterial adhesion strategy.  The majority of the concepts are connected to in vitro studies. I would add to the outlook section or in a separate heading a small paragraphs on what are the current in vivo studies on the application of the mentioned strategy. Are there any randomized control trials or other in vivo studies? or the research is limited to in vitro article?

Additionally I would suggest to add an other little paragraph on the strategies to treat peri-implantitis.

A very recent article that match your paper topic was published, I encourage you to add it to the Manuscript to help the reference improvement.

Alovisi, M.; Carossa, M.; Mandras, N.; Roana, J.; Costalonga, M.; Cavallo, L.; Pira, E.; Putzu, M.G.; Bosio, D.; Roato, I.; Mussano, F.; Scotti, N. Disinfection and Biocompatibility of Titanium Surfaces Treated with Glycine Powder Airflow and Triple Antibiotic Mixture: An In Vitro Study. Materials 202215, 4850. https://doi.org/10.3390/ma15144850

Author Response

Response to Reviewer 2 Comments

Point 1:Abstract : ''Dental implants are widely used to restore missing teeth because of their stability, comfort,  and aesthetic characteristics''.  Aesthetic characteristics are not linked to dental implants, but rather with the crown that rehabilitates the implant. Please reformulated to avoid misunderstanding. Same with the first phrase of the introduction.

Response 1:Thanks for your comment. The aesthetics characteristics are mainly linked to the crown, and dental inmlants are widely used due to their stability and comfort. We have deleted the description of aesthetics in Line 11, 24.

Point 2:The article well covers the aspects of the bacteria colonization and the anti bacterial adhesion strategy.  The majority of the concepts are connected to in vitro studies. I would add to the outlook section or in a separate heading a small paragraphs on what are the current in vivo studies on the application of the mentioned strategy. Are there any randomized control trials or other in vivo studies? or the research is limited to in vitro article?

Additionally I would suggest to add an other little paragraph on the strategies to treat peri-implantitis.

Response 2:Thanks for your suggestion. We are sorry for the neglect of in vivo studies. They are important for the development of antibacterial adhesion strategy for dental titanium implant surface.We have paid more attention about in vivo studies and searched more randomized control trials. We have added the current in vivo studies on the strategies to treat peri-implantitis in Line 411-418.

Point 3:A very recent article that match your paper topic was published, I encourage you to add it to the Manuscript to help the reference improvement.

Alovisi, M.; Carossa, M.; Mandras, N.; Roana, J.; Costalonga, M.; Cavallo, L.; Pira, E.; Putzu, M.G.; Bosio, D.; Roato, I.; Mussano, F.; Scotti, N. Disinfection and Biocompatibility of Titanium Surfaces Treated with Glycine Powder Airflow and Triple Antibiotic Mixture: An In Vitro Study. Materials 2022, 15, 4850. https://doi.org/10.3390/ma15144850

Response 3:Thanks for your suggestions. This article is closely matched our manuscript. It makes my manuscript more complete by studies of other treatments. We have added this reference numbered 32 in Line 61. 

Reviewer 3 Report

The authors submitted a review with the title: "Antibacterial Adhesion Strategy for Dental Titanium Implant Surface: From Mechanisms to Application".
The article is written comprehensibly, logically divided. 

In the Introduction, they describe the indications and contraindications of dental implants, formation and prevention of biofilm.
In other chapters, the authors deal with the adhesion process of oral bacteria, implant surface properties affecting bacterial adhesion and anti-adhesion strategies for titanium implnats.
Review contains 168 sources of references, which implies that the authors have sufficiently devoted themselves to the given issue. 

It is also visible in the professional processing and presentation of information in the review.
I have no reservations about the article, congratulations to the author for a great job.

Author Response

Response to Reviewer 3 Comments

Thank you very much for your careful review and affirm with regard to our manuscript #JFB-1938526, entitled “Antibacterial Adhesion Strategy for Dental Titanium Implant Surface: From Mechanisms to Application”. We appreciate for your warm work earnestly. We will continue to work in this field. Please continue to pay qttation to our search.

Thank you again for your attention to our manuscript.

Round 2

Reviewer 1 Report

The authors have revised the manuscript properly. I have no further comments.

Author Response

Response to Reviewer 1 Comments

Thank you very much for your careful re-review and affirm with regard to our manuscript #JFB-1938526, entitled “Antibacterial Adhesion Strategy for Dental Titanium Implant Surface: From Mechanisms to Application”. We appreciate for Editors/Reviewers’ warm work earnestly. We will continue to work in this field. Please continue to pay qttation to our search.

Thank you again for your attention to our manuscript.

Reviewer 2 Report

Dear Authors,

Thank you for addressing my comments. The article is of interest and i consider it complete and worthy to be published.

Author Response

Response to Reviewer 2 Comments

Thank you very much for your careful re-review and affirm with regard to our manuscript #JFB-1938526, entitled “Antibacterial Adhesion Strategy for Dental Titanium Implant Surface: From Mechanisms to Application”. We appreciate for Editors/Reviewers’ warm work earnestly. We will continue to work in this field. Thanks for yor interest and please continue to pay qttation to our search.

Thank you again for your attention to our manuscript.